# Graph Reordering for Cache-Efficient Near Neighbor Search

**Benjamin Coleman**[*]
ECE Department
Rice University
Houston, TX 77005
benjamin.ray.coleman@gmail.com

**Santiago Segarra**
ECE Department
Rice University
Houston, TX 77005
segarra@rice.edu

**Alex Smola**
Amazon Web Services
alex@smola.org

**Anshumali Shrivastava**
Department of Computer Science
Rice University
Houston, TX 77005
anshumali@rice.edu

## Abstract

Graph search is one of the most successful algorithmic trends in near neighbor search. Several of the most popular and empirically successful algorithms are, at their core, a greedy walk along a pruned near neighbor graph. However, graph traversal applications often suffer from poor memory access patterns, and near neighbor search is no exception to this rule. Our measurements show that popular search indices such as the hierarchical navigable small-world graph (HNSW) can have poor cache miss performance. To address this issue, we formulate the graph traversal problem as a cache hit maximization task and propose multiple graph reordering as a solution. Graph reordering is a memory layout optimization that groups commonly-accessed nodes together in memory. We mathematically formalize the connection between the graph layout and the cache complexity of search. We present exhaustive experiments applying several reordering algorithms to a leading graph-based near neighbor method based on the HNSW index. We find that reordering improves the query time by up to 40%, we present analysis and improvements for existing graph layout methods, and we demonstrate that the time needed to reorder the graph is negligible compared to the time required to construct the index.

## 1 Introduction

Near neighbor search is a fundamental building block within many applications in machine learning systems. Informally, the task can be understood as follows. Given a dataset $D = \{x_1, x_2, ...x_N\}$, we wish to build a *data structure* that can be queried with any point $q$ to obtain the $k$ points $x_i \in D$ that have the smallest distance to the query. This structure is called a *near neighbor index*. Near neighbor indices are of tremendous practical importance, as they form the backbone of production models in recommendation systems, natural language processing Mikolov et al. (2013), genomics Ondov et al. (2016), computer vision Lowe (1999) and other applications in machine learning Shakhnarovich et al. (2008).

**Applications.** The accuracy and speed of several machine learning algorithms critically depend on the recall and latency of near neighbor search. The problem is the focus of intense research activity

---

[*]Work was completed while the author was with Amazon Web Services.

36th Conference on Neural Information Processing Systems (NeurIPS 2022).

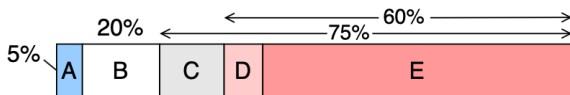

Figure 1: Breakdown of query subroutines: initialization (A), visited lists (B), node traversal (C), and distance computation (D-E): $\approx 40\%$ of the query time is spent accessing vectors from memory (E). The remainder (D) is floating point math.

due to its central role in neighbor-based classification, regression and prediction with embedding models. In a neural embedding model, objects are transformed into *embeddings* in $\mathbb{R}^d$, where $d$ ranges from 100 to 1000 and $N$ often exceeds 100 million. Prediction typically consists of finding the $k$ nearest embeddings. For example, Amazon's deep semantic search engine works by embedding the product catalog and the search query into the same high-dimensional space. The engine recommends the products whose embeddings are nearest to the embedded search query Nigam et al. (2019).

Since the search occurs for every query, the latency and recall of the system depend on the ability to perform fast search in the high-recall regime. Similar problems exist in natural language processing Kusner et al. (2015), information retrieval Huang et al. (2013), computer vision Frome et al. (2013) and other application domains. Beyond machine learning, near neighbor indices are used to match audio recordings, detect plagiarism, classify ECG signals in healthcare, and perform copyright attribution with blockchain Jafari et al. (2021). The problem is a central bottleneck for many applications.

**Latency Challenges.** Solutions to the near neighbor problem are incredibly diverse, ranging from hardware-accelerated brute force Johnson et al. (2019) to space-partitioning trees Beygelzimer et al. (2006). Graph algorithms have emerged as a markedly effective class of methods for high-dimensional near neighbor search. For example, four of the top five search libraries on the well-established ANN-benchmarks leader board use a graph index Aumüller et al. (2020). Because these libraries have been integrated with many large-scale production systems Johnson et al. (2017); Malkov & Yashunin (2018), they have been hand-optimized in C++ at a large engineering cost. Practical techniques for fast graph search represent a highly active research area with clear impact.

**Profiling the Query Time of HNSW.** Figure 1 shows the relative time spent performing different operations in the hierarchical navigable small-world graph (HNSW) algorithm, which we obtained using the *perf* tool De Melo (2010) on the leading nmslib implementation. Approximately 40% of the time is spent on instructions corresponding to the first dimension of the distance computation, which is dominated by the cost of fetching the vectors from main memory. Thus, a good strategy to reduce the query latency is to target its major bottleneck, which appears to be memory fetching.

## 1.1 Our Contribution

In this paper, we propose *graph reordering* to improve approximate near neighbor search. Graph reordering is a cache optimization that places neighboring nodes in consecutive (or near-consecutive) memory locations. The new memory layout can speed up traversal-based graph algorithms that are constrained by RAM access. The mechanism commonly understood to drive performance – that of packing multiple nodes into a cache line – does not apply to near neighbor graphs, where a single node spans many cache lines. However, we demonstrate that hardware prefetching mechanisms make reordering a viable optimization even in this large-node setting. Our work reveals that near neighbor search can be significantly accelerated through graph reordering, a hiterto unexplored optimization for this important task.

**New Reordering Results.** Graph reordering has not been theoretically analyzed under any formal computational model. Beginning from first principles, we analyze the cost of breadth-first search under the idealized cache model for several layouts and we develop an objective to find an optimal-cost layout. We also introduce *profile-guided* reordering, an optimization that uses the empirical frequency of edge traversal to inform the co-location of nodes. By including observations of query traffic patterns through the graph, we obtain a speedup over query-agnostic algorithms. We show that the problem reduces to *weighted* graph reordering, which we solve with a greedy algorithm.

**Extensive Experimental Benchmark.** We present a rigorous evaluation where we integrate graph reordering into HNSW. We perform an exhaustive comparison on large embedding datasets where we benchmark the query time and cache miss rate. Our experiments show that graph reordering

is a viable and effective way to improve practical machine learning deployments that involve near neighbor search. We make the following new observations:

**1. Latency Improvement:** Graph reordering improves the mean and 99[th] percentile latency by up to 40% and 20%, respectively.

**2. Small Marginal Cost:** The reordering time is an order of magnitude smaller than the time needed to construct the search index.

**3. Query-Enhanced Ordering:** Our profile-guided method provides an additional speedup over existing query-agnostic algorithms.

**4. Theoretical Analysis:** We rigorously analyze popular heuristics in the ideal cache model, showing that objective-based methods provably attain speedups (while lightweight methods do not).

## 2 Background and Related Work

**CPU Cache Hierarchy.** Modern computers organize memory access into a series of intermediate storage locations known as *caches*. When the CPU core is instructed to load data from RAM, it first checks to see whether the data resides in the nearest (L1) cache. If there is a *cache miss* (i.e. the data is not found), the processor performs a secondary search in the L2 and L3 caches, resorting to main memory if the data is not present in any cache. Caches are organized into sets of *cache lines*, or small (32 or 64 byte) blocks of memory that are always moved together. Upon a cache miss, the CPU evicts the least recently-used line to make room for the requested line. The access times for L1, L2, L3 and DRAM are about 1, 3, 10, and 100 nanoseconds, respectively Levinthal (2009). To ensure that data resides in the nearest possible cache, one may repeatedly access the same data (temporal locality) or ensure that sequentially-accessed data have nearby memory addresses (spatial locality).

**Hardware and Software Prefetching.** Cache lines are smaller than the size of many data objects. To increase cache hits, modern CPUs use a *prefetcher* to predict future lookups and preemptively place the corresponding lines in the L1 or L2 cache. Hardware prefetching is done by the processor using heuristics and simple learning algorithms. Software prefetching happens when the programmer manually inserts special prefetch instructions into the code. The two mechanisms interact in complex ways, with both synergistic and antagonistic effects, but improving the spatial locality of memory access is a sound principle to improve performance Lee et al. (2012).

**Near Neighbor Graphs.** Graph-based algorithms such as HNSW, pruned approximate near neighbor graphs (PANNG) Iwasaki (2016), and optimized near neighbor graphs (ONNG) Iwasaki & Miyazaki (2018) feature prominently in leader boards and production systems at web-scale companies. Apart from recent theoretical progress Prokhorenkova & Shekhovtsov (2020), the primary focus of graph index research has been to develop heuristics (such as diversification, pruning, and hierarchical structures) that improve the properties of the graph, increase search accuracy, and reduce search time. We argue that most search algorithms have essentially the same computational workload because these heuristics produce highly similar graph structures.

**Graph Structure and Graph Search.** The core algorithmic components of graph search remain the same across most popular graph indices. We begin with a graph constructed by connecting each point in the dataset to a set of nearby points. These edges undergo diversification and pruning, where edges are added/deleted based on method-dependent heurstics. To perform the search, we select an initial node and walk along the edges of the graph until a neighbor is found, recording the distance between the node and query at each step. Our crucial observation is that major search indices do not differ in terms of node access pattern or graph properties relevant to reordering. We provide a more complete description in the supplementary materials.

**Cache Locality of Graphs.** Due to their industrial importance, search index implementations have been optimized aggressively in terms of memory access and computation. For example, the HNSW software manual Naidan et al. (2015) describes a "flattened index" to improve the memory access pattern and reduce memory fragmentation Boytsov & Naidan (2013). The flat layout stores nodes and data together in a contiguous pre-allocated block of memory. By avoiding pointers, the efficient layout avoids unnecessary implicit addressing and improves the ability of software and hardware prefetching to improve the algorithm Malkov & Yashunin (2018). The method provides sound improvements, but it leaves some performance on the table because it is agnostic to the graph structure. Graph reordering

is an orthogonal improvement that uses information about the graph to further improve the layout by re-ordering nodes *within the contiguous block of memory*.

**Ordering with Space-Filling Curves.** Another similar idea involves the use of space-filling curves for search problems Chan (2002). A space-filling curve is the shortest line that passes through each point in the dataset. One may use a cache efficient linear scan through the ordered data for fast search and graph construction in low-dimensional ($d \leq 3$) spaces Connor & Kumar (2010). However, the method suffers from the curse of dimensionality and does not work well for machine learning problems. To the best of our knowledge, our paper is the only investigation of graph reordering for fast near neighbor queries.

# 3   Graph Reordering

Formally, graph reordering can be seen as constructing a labeling function $P : V \rightarrow \{1, \ldots, N\}$ that assigns each node $v \in V$ in a graph to a unique integer index (or label) between 1 and $N$ following a pre-specified rule or in order to maximize some objective. Many formulations are possible, but we typically expect $P$ to map connected nodes to similar (nearby) labels. The function $P$ is then used as the memory layout for the graph, with node $v$ assigned to memory location $P(v)$.

**Existing Techniques for Graph Reordering.** We assess the ability of several recent methods to accelerate near neighbor search. We consider the reverse Cuthill-McKee order (RCM) Cuthill & McKee (1969); George (1971), gorder Wei et al. (2016), degree sorting, hub sorting Zhang et al. (2017), hub clustering Balaji & Lucia (2018), and degree-based grouping Faldu et al. (2019). Gorder and RCM are objective-driven methods that seek to a minimizer $P$ of a cost function, while the others are degree-based heuristics.

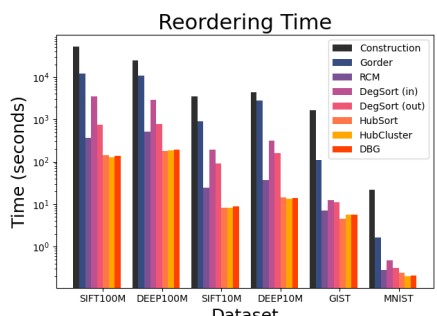

**Gorder.** We discuss all layout methods in the supplementary materials, but Gorder (graph-order) is the current state of the art to accelerate analyses such as PageRank and graph diameter Wei et al. (2016). The Gorder layout $P_{\mathrm{GO}}$ optimizes the average neighborhood overlap between the connections of nearby nodes by maximizing the number of shared edges among size-$w$ blocks of consecutive nodes. Since nodes are penalized for having neighbors further than $w$ locations away, this objective is a reasonable proxy for cache efficiency.

Figure 2: In our experiments, the reordering algorithms required less time to run than graph construction.

$$P_{\mathrm{GO}} = \arg\max_P \sum_{\substack{u,v \in V \,\mathrm{s.t.} \\ |P(u)-P(v)|<w}} S_s(u,v) + S_n(u,v) \qquad (1)$$

We use the notation $S_s(u,v)$ to count the number of directed connections between $u$ and $v$, $S_n(u,v)$ to count the number of shared in-neighbors between $u$ and $v$, and $F_{\mathrm{GO}}(P)$ for the objective score. Two consecutive nodes improve $F_{\mathrm{GO}}(P)$ if they share a direct edge or have many common neighbors. Maximizing the objective is NP-hard, but a greedy algorithm provides a $1/(2w)$-approximate solution.

# 4   Theory

Here, we formalize the connection between graph layout and cache performance. We defer all proofs to the supplementary materials.

**Idealized Cache Model.** Our main theoretical contribution is to analyze the *cache complexity* of graph search under the idealized cache model Frigo et al. (1999) for various graph layouts. The idealized cache model is a theoretical model of computation that counts the number of cache misses. The model assumes a cache with $B$ "objects" per line and $T$ cache lines, which are evicted in the order that minimizes the overall number of cache misses.

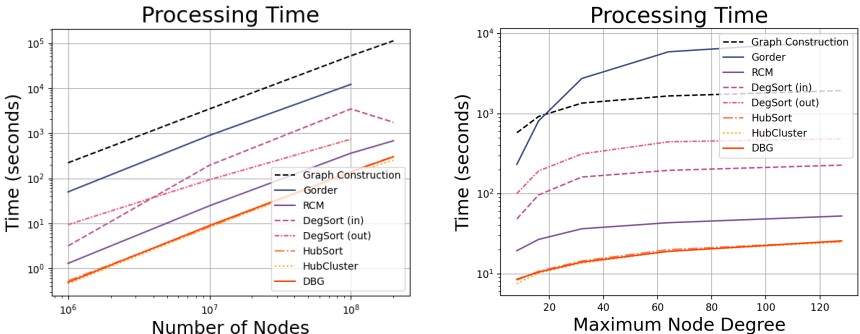

Figure 3: Graph reordering scales to large datasets and complex near neighbor graphs. In our experiments, the reordering algorithms required less time to run than graph construction. Reordering is feasible even for large graphs with many nodes (SIFT with $k = 16$, middle) and for densely connected graphs ($k \leq 120$) on large datasets (GIST1M, right).

Even though the size of a graph node exceeds the size of a real cache line, we suppose that each line contains $B$ nodes. This is reasonable because next-line prefetchers often implement this behavior and because auxiliary data structures, such as visited lists, represent nodes using only a few bytes. We also assume that the cache is empty at the start of the program, and we analyze the cost to perform one step of breadth-first search under the common "tall cache" assumption (i.e. $T > B^2$).

It is important to note that our analysis is pessimistic in the sense that the cache might include neighbors from previously-accessed nodes, which could reduce the cost to explore a neighborhood. While real-world performance does benefit from this situation, our one-step clean-cache analysis provides a lower bound on the performance.

Given a graph $G = (V, E)$, we consider the number of cache misses, or the cost $C_i$, of accessing a node $v_i$ and its neighbors. We also consider the *cache efficiency* $\mathrm{CE}_i$, or the number of cache hits involved when accessing a node and then each of its out-neighbors in sequence.

$$C_i = 1 + \sum_{\substack{v_j \in E(v_i) \\ j=1\ldots\deg(v_i)}} \mathbb{1}_{\{v_j \notin l(v_i)\}} \prod_{k<j} \mathbb{1}_{\{v_j \notin l(v_k)\}} \qquad \mathrm{CE}_i = \sum_{\substack{v_j \in E(v_i) \\ j=1\ldots\deg(v_i)}} \mathbb{1}_{\{v_j \in l(v_i) \bigcup_{k<j} l(v_k)\}}$$

Here, $l(v)$ is the cache line containing $v$ and $E(v)$ is the set of outbound edges of $v$. The 1 represents the initial cost to load $v_i$, and the sum represents the cost to load the lines containing neighbors of $v_i$.

**Goal of Reordering.** We seek to minimize the average cost of depth-first search over the graph, or equivalently maximize the average efficiency $\mathrm{CE}_{\mathrm{DFS}} = \frac{1}{N} \sum_{v_i \in V} \mathrm{CE}_i$. That is, we wish to find $P^\star = \arg\max_P \mathrm{CE}_{\mathrm{DFS}}(P)$.

**Cache Complexity Objective.** The cache efficiency score is cumbersome to work with directly. We use the following inequalities to transform $\mathrm{CE}_{\mathrm{DFS}}(P)$ into a form that is more amenable to optimization and analysis.

**Theorem 1.** *Given a graph $G = (V, E)$, let*

$$\mathrm{UB} = \sum_{i=1}^{N} \left( \sum_{j=\lfloor i/B \rfloor + 1}^{i-1} S_s(v_i, v_j) + S_n(v_i, v_j) - \max\left\{ \sum_{j=\lfloor i/B \rfloor + 1}^{\lceil i/B \rceil} \mathbb{1}_{\{(v_i, v_j) \in E\}} - 1, 0 \right\} \right)$$

*Then the cache efficiency score $\mathrm{CE}_{\mathrm{DFS}}(P)$ obeys the following inequalities.*

$$\mathrm{UB} - \sum_{i=1}^{N} \mathrm{Triplets}(v_i) \leq \mathrm{CE}_{\mathrm{DFS}}(P) \leq \mathrm{UB}$$

*where $S_s(v_i, v_j)$ and $S_n(v_i, v_j)$ are as defined in Equation 1 and $\mathrm{Triplets}(v_i)$ is the number of triplet combinations of $v_i$'s out-neighbors that fall into the same cache line.*

Note that when $B \leq 2$, the upper bound and lower bound are equal (since we cannot form triplets from $B = 2$ items). It is easy to show that this problem is an NP-hard instance of the maximum weight perfect hypergraph matching problem.

**Lemma 1.** *Given a graph $G = (V, E)$, the problem of finding the layout with an optimal DFS cache cost is an NP-hard instance of maximum weight perfect hypergraph matching.*

**Analysis of Gorder.** One interesting and new observation is that the Gorder objective (Equation 1) is similar to our upper bound on cache complexity. By bounding the size of the "extra" terms in the upper and lower bounds, we can analyze the cost of the layout found by Gorder.[2]

**Theorem 2.** *Let $G = (V, E)$ be a graph with a maximum out-degree $M$, $P$ be a layout with objective score $F_{\mathrm{GO}}(P)$, and $P'$ be the layout obtained by running Gorder plus a simple cache boundary alignment procedure on $P$. $P'$ satisfies the following inequalities:*

$$\mathrm{CE}_{\mathrm{DFS}}(P') \geq \frac{1}{B} F_{\mathrm{GO}}(P) - N(B-1) - N\lfloor M/B \rfloor \binom{B}{3} \qquad \mathrm{CE}_{\mathrm{DFS}}(P') \leq F_{\mathrm{GO}}(P)$$

*We use the convention $\binom{a}{b} = 0$ when $a < b$.*

Using these inequalities, we can derive conditions under which Gorder is guaranteed to improve the cache efficiency.

**Theorem 3.** *Given two node orderings $P_1$ and $P_2$ on a graph with maximum out-degree $M$, $\mathrm{CE}_{\mathrm{DFS}}(P_2) \geq \mathrm{CE}_{\mathrm{DFS}}(P_1)$ if $F_{\mathrm{GO}}(P_2) \geq BF_{\mathrm{GO}}(P_1) + B(B-1)N + NM\binom{B}{3}$.*

Theorem 3 guarantees a cache efficiency improvement if we increase the Gorder objective (and run a simple alignment process). However, the objective must improve substantially for the theory to apply: the final score should be $\geq B$ times the initial score, not to mention the combinatorial $B$ term and linear $N$ term. To address this concern, we computed the initial and final scores after running Gorder with $B = 2$ (Table 2). We found that the improvement is large enough to guarantee cache benefits on practical tasks.

**Implications for Other Methods.** Lightweight algorithms cannot provably decrease the cache complexity without a power-law assumption about the graph. For a simple counterexample, consider a nearly-regular graph where all nodes have the same degree, except one. Degree-based sorting can easily move this node away from its neighbors, increasing the cost. A similar argument holds for the RCM order, since it considers only the maximum label discrepancy.

## 4.1 Improving the Ordering Process

Theorem 2 requires a combinatorial term to bound the deviation of $\mathrm{CE}_{\mathrm{DFS}}(P)$ from its upper bound. To investigate the gap between this bound and existing methods, we examine a "Corder" objective (for cache order) which directly maximizes the lower bound. We also introduce a method based on empirical profiling of the node access pattern, which we call "Porder" (for profile order).

**Cache Order.** The Corder layout is obtained as follows:

$$P_{\mathrm{CO}} = \arg\max_P F_{GO}(P) - \sum_{\substack{u,v,m \in V \text{ s.t.} \\ |P(a)-P(b)|<w \\ \forall (a,b) \in \{u,v,m\}}} S_t(u, v, m) \tag{2}$$

$S_t(u, v, m)$ is the number of shared parents among the triplet $(u, v, m)$. Thanks to this additional term, we optimize for the lower bound of Theorem 1 rather than the upper bound. We use a greedy selection algorithm to solve the optimization problem.

**Profile Order.** Intuition suggests that some parts of the search index graph are visited more frequently than others. Our Porder algorithm prioritizes important neighborhoods by considering a *weighted* graph order problem. We provide pseudocode in the supplementary materials and an application-agnostic implementation in the code.

---

[2]Plus a cache alignment procedure described in the supplementary materials.

Table 1: Perf stat results for cache misses and TLB misses on SIFT100M. Other datasets show similar results.

|          | Original | RCM   | Gorder |
| -------- | -------- | ----- | ------ |
| L1 (%)   | 19.53    | 17.37 | **14.46** |
| L2 (%)   | 13.9     | **7.61** | 9.6 |
| L3 (%)   | 6.5      | 5.1   | **4.0** |
| TLB (%)  | 3.85     | 2.56  | **2.14** |

Table 2: Initial ($F_i$) and final ($F_f$) Gorder scores with $B = 2$. $M$ is the maximum out-degree.

| Dataset  | $M$ | $N$  | $F_i$ | $F_f$ |
| -------- | --- | ---- | ----- | ----- |
| SIFT1M   | 8   | 1M   | 6.6k  | 2.6M  |
| F-MNIST  | 4   | 60k  | 64    | 102k  |
| NYTimes  | 8   | 290k | 417   | 736k  |

Table 3: Cache miss rates for node traversals and distance computations on SIFT100M (lower is better). It should be noted that due to the 100x latency difference between cache and RAM, small improvements to the cache miss rate can substantially speed up an algorithm. The ranking of algorithms by cache miss rate agrees with the ranking by speedup in Figure 4.

|        | Original | Gorder | RCM   | DegSort (in) | DegSort (out) | HubSort | HubCluster | DBG   |
| ------ | -------- | ------ | ----- | ------------ | ------------- | ------- | ---------- | ----- |
| L1 (%) | 22.76    | 19.28  | 20.61 | 22.76        | 22.76         | 22.85   | 22.77      | 22.81 |
| L3 (%) | 13.56    | 8.32   | 8.91  | 13.55        | 13.56         | 13.63   | 13.51      | 13.34 |

Each edge is weighted proportional to its importance for near neighbor queries. In practice, we estimate the edge weights by issuing a small number of search queries and counting the number of times the edge is traversed in the query process. This idea may be incorporated into many existing ordering techniques, but we consider a weighted version of Equation 1.

$$P_{\text{PO}} = \arg\max_{P} \sum_{\substack{u,v \in V \text{ s.t.} \\ |P(u)-P(v)|<w}} S_{ws}(u,v) + S_{wn}(u,v), \tag{3}$$

Here, $S_{ws}(u,v)$ computes the sum of weighted out-edges between $u$ and $v$ and $S_{wn}(u,v)$ computes the sum of weighted in-edges from neighbors shared by $u$ and $v$.

## 5  Experiments

**Experiment Setup.** Our experiments measure index performance using wall-clock query time with different memory layouts.

**Query Latency.** We benchmark the system after a "warm start," where assets such as the index, query, and data are pre-loaded into RAM. To avoid the difficulties associated with timing very short events, we record the total time needed to perform 10k queries and report the average query time. Finally, we restrict the query program to a single core and ensure that the benchmark is the only program running on the server. We run all of our experiments on a server with 252 GB of RAM, 28 Intel Xeon E5-2697 CPUs, and a shared 36 MB L3 cache.

**Cache Misses.** We measure cache miss rates using the Linux perf tool to record hardware CPU counters for events such as data reads and cache hits. We use these counters to compute the cache miss rate by dividing the number of data cache misses by the number of data references. We run our most successful reordering methods (Gorder and RCM) on SIFT100M and report results for the L1, L2, L3 and TLB caches. To validate these measurements, we also run the program through the cachegrind virtual processor. Using cachegrind, we annotate the source code of our program to verify that the cache miss reduction is from lines of code that execute node traversals and distance computations.

**Implementation.** We extended the nmslib implementation of HNSW to support graph reordering, with minor changes to speed up graph construction. We use the flat layout described by Naidan et al. (2015) to avoid memory fragmentation. To ensure that we benchmark a realistic workload, we verified that our implementation uses the same memory layout, requires the same number of distance computations and produces the same graph as nmslib-HNSW. We implement eight reordering

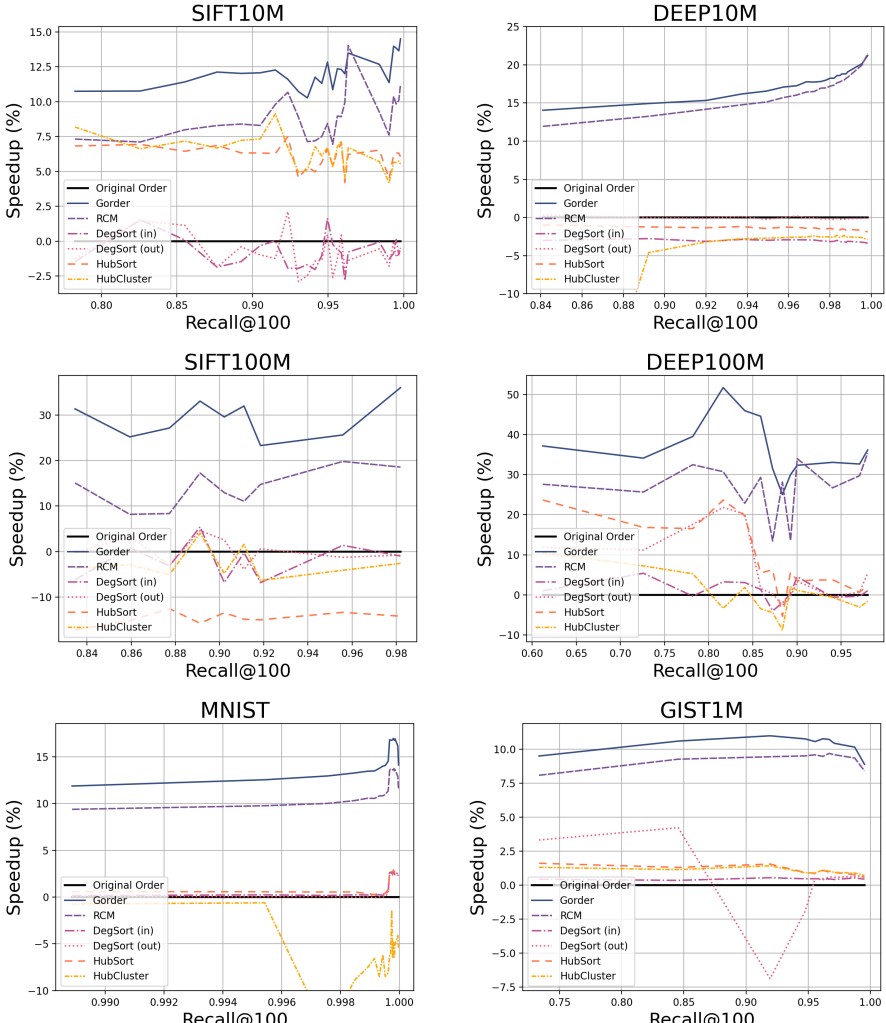

Figure 4: Effect of graph reordering on the query time vs. recall tradeoff. Reordering algorithms above the black line have faster query time than the original ordering. Algorithms below the line cause a slowdown. The speedup changes with the recall because the memory access pattern of beam search changes with the buffer size.

methods, including our techniques from Section 4. We do not use graph partitioning because such methods generate orders that are too coarse to accelerate cache access Lecuyer & Tabourier (2021). Additional details about baseline selection and memory layout are available in the supplementary materials and code documentation.

**Datasets.** We use large datasets that are representative of embedding search tasks. We perform experiments on datasets from ANN-benchmarks as well as on 10M and 100M sized subsets of the SIFT1B and DEEP1B benchmark tasks. Table 4 contains information about our datasets. In all of our experiments, we request the top 100 neighbors of a query and we report the recall of the top 100 ground truth neighbors.

**Hyperparameters.** For construction, HNSW requires two parameters: the maximum number of edges for each node ($k_c$) and the size of the beam search buffer ($M_c$) used to find the $k_c$ neighbors during graph construction. We set $M_c = 100$ and construct indices for $k_c \in \{4, 8, 16, 32, 64, 96\}$. From these options, we select the index with the best recall-latency trade-off (for recall $> 0.95$) and use that index for our graph reordering experiments. While it is true that the best hyperparameters change based on the recall, there is typically a clear winner for the high-recall regime. To query the index, HNSW requires the beam search buffer size ($M_q$), which controls the trade-off between recall

Table 4: Each dataset has $N$ entries and $d$ features ("vector size" bytes / row).

| Dataset | $N$ | $d$ | vector size |
|---------|------|-----|-------------|
| GIST | 1 M | 960 | 3.8 kB |
| SIFT | 10 - 100 M | 128 | 128 B |
| DEEP | 10 - 100 M | 96 | 384 B |
| MNIST | 60 k | 784 | 3.1 kB |

Table 5: Query time ($T_Q$) and speedup on SIFT10M and Yandex 1B T2I.

| Algorithm | $T_Q$ (Speedup %) | |
|-----------|---------|-----------|
| | SIFT10M | Yandex 10M |
| None | 22.8 (0%) | 16.3 (0%) |
| Gorder | 17.9 (27%) | 12.8 (22%) |
| Porder | **16.8 (36%)** | 12.4 (24%) |

and query latency. We reorder the graph and issue the same set of 10K queries for each ordering. When we query the index, we vary the beam search buffer size $M_q$ from 100 to 5000. We use window size $w = 5$ for Gorder and use the standard hyperparameter settings suggested by the literature for the other methods. For our Porder method, we create the weighted graph by profiling on the first 1K queries from the test set and reporting results over 10K test queries.

# 6 Results

**Query Time.** Figure 4 shows the effect of graph reordering on the query time vs. recall tradeoff. We report the R100@100 recall, or the recall of the top 100 ground truth neighbors among the top 100 returned search results. For a given recall value, we calculate the speedup as the ratio of average latency without/with reordering, where the averages are over 10K queries and 5 runs. Finally, we measured the 99[th] percentile of latency (P99) for the SIFT100M dataset in Figure 5, to ensure that our altered graph layout does not increase the tail latency. Gorder and RCM improve the P99 latency by at least 20% in the high-recall regime.

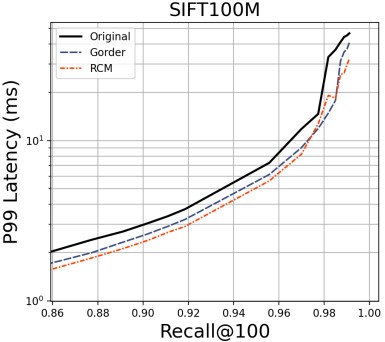

Figure 5: 99[th] percentile of latency (P99). RCM (-17%) and Gorder (-30%) outperform the original order.

**New Layout Methods.** We conducted a separate study to determine whether profile-guided reordering (Porder) can further refine the layout obtained using existing techniques. We construct a weighted graph by issuing 1000 queries and counting the number of times $T_e$ that edge $e$ is traversed by the search algorithm. The edges are weighted as $1 + T_e$ and used in Equation 3. We found that Porder improved over Gorder by up to 5-10% (Table 5). We also implemented Corder as an ablation study to determine whether there is a practical performance gap between the upper bound (Gorder objective) and lower bound (Corder objective) of Theorem 1. On SIFT10M, we found that Corder was 4% *worse than Gorder*, leading us to conclude that Gorder obtains a layout with near-optimal cache efficiency (i.e. the combinatorial term is small).

**Reordering Cost and Cache Performance.** Figure 3 presents the cost of reordering the graph. Reordering time scales well with the number of nodes $N$ and the maximum degree $k_c$ of each node. Table 1 contains perf cache measurements and Table 3 contains cachegrind information for graph search over the SIFT100M dataset. Because these two experiments have similar results, we conclude that the mechanism by which Gorder and RCM improve latency is by reducing cache miss rate.

**Prefetcher and Node Size Ablations.** To isolate the effect of cache coherency from that of hardware prefetching, we turned off the L1/L2 hardware prefetcher. Without L1/L2 prefetching (but with TLB caching and L3 prefetch), we obtain speedups that are much smaller than when the prefetcher is active (-10% on MNIST, -5% on DEEP10M and -5% on SIFT10M). We also consider two equivalent versions of the SIFT1M dataset - one where vectors are represented as 32-bit floats (512 bytes / vector) and one where vectors are represented as 8-bit integers (128 bytes / vector). We use the same graph for each index and observe that larger vectors result in a smaller speedup (average of 11% vs 19%). These results support the idea that prefetching and vector size play a substantial role in our latency improvement. See the supplementary material for detailed experiment results.

# 7 Discussion

Objective-based reordering algorithms such as Gorder, Porder and RCM are the most effective methods to accelerate near neighbor search, with a typical speedup of 10% on small datasets ($N < 1$M) and speedups of up to 40% on large datasets and in the high-recall regime (Figure 4). Our theoretical and experimental results suggest that graph reordering should become a standard preprocessing step to improve the query time. In practice, reordering only affects the representation of nodes in memory and does not change the recall, search algorithm, or other properties of the graph index. Furthermore, reordering does not substantially inflate the index construction time and, using the theory from Section 4, we argue that Gorder does no harm under typical application scenarios.

**Cost of Reordering.** Many studies focus on applications such as PageRank, where the graph is processed a small number of times to obtain an output. Such applications favor lightweight (but less effective) algorithms that obtain an *end-to-end* speedup. However, the production requirements of near neighbor search are exactly the opposite: a single index may be queried millions of times over its life cycle. Therefore, the reordering cost is justified, especially since it is an order of magnitude smaller than HNSW construction for most datasets (Figure 3).

**Benefits of Reordering.** Reordering is an index-agnostic method to improve the performance of graph-based near neighbor search. Because reordering depends on the node access pattern of beam search, which is common to most algorithms, reordering is applicable to a wide range of practical search tasks. It should be noted that most real-world deployments function under strict latency requirements: a maximum search time of 20 ms is a common constraint Nigam et al. (2019). A 20% improvement in search time allows the system to perform more sophisticated processing of the search results, operate at a higher recall or serve a larger collection of embeddings.

**What drives performance?** A natural question is to ask what mechanisms drive the performance improvements seen in our experiments. Since data vectors are larger than a single cache line, prefetching is likely responsible for many of the cache improvements seen in Table 1. Reordering the graph improves the spatial locality of nodes, allowing the hardware prefetcher to be more effective, as well as auxiliary data structures, such as visited lists and priority queues. Although auxiliary functions are only responsible for about 20% of the query time (Figure 1), they represent nodes using only a few bytes and therefore *do* fit into a single cache line. These structures benefit from concurrent loading as well as prefetching.

**Limitations.** It is possible that a highly-engineered implementation with software prefetching could have similar performance to a naive implementation that uses reordering. However, we believe that graph reordering is valuable even for a tuned implementation. Studies show that both hardware and software prefetching work best on data that obeys the spatial locality principle Lee et al. (2012), and software prefetching comes at a high engineering cost.

**Synergy with Algorithms and Production Systems.** Reordering is likely to exhibit synergy with recent $k$-NN algorithms. For example, partition-based search may benefit from reordering because recent algorithms for locality-sensitive hashing use $k$-NN graph cuts to form data partitions Dong et al. (2020). Near neighbor graphs are also frequently combined with sample compression methods such as product quantization or other codebooks Jegou et al. (2010). Recent experiments with preprocessing transformations suggest that it is beneficial to perform the graph search on a subspace of reduced dimension Prokhorenkova & Shekhovtsov (2020). This could amplify the effects of graph reordering because a larger sub-graph can fit into the CPU cache for small vector sizes. Our experiments support this hypothesis, as we observe larger speedups for SIFT and DEEP than MNIST and GIST, which have a large per-sample storage cost. Finally, we expect our profile-guided technique to be beneficial for near neighbor indices in production systems. It is common for production databases to periodically re-format the contents of the database according to recent statistics about access frequency and workload. Since reordering the index only requires a few minutes, we find it useful to dynamically adjust the memory layout when the query distribution changes.

# Acknowledgements

This work was supported by National Science Foundation SHF-2211815, BIGDATA-1838177, ONR DURIP Grant, and grants from Adobe, Intel, Total, and VMware. We also thank Senthil Rajasekaran for help with the formulation and proof of Lemma 1.

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
