# OpenReview forum: "Graph Reordering for Cache-Efficient Near Neighbor Search"
_NeurIPS.cc/2022/Conference — NeurIPS 2022 Accept_

### Official Review · Reviewer_Ewkj · 2022-06-28

**Rating:** 6
**Confidence:** 4
**Soundness:** 3 good
**Presentation:** 3 good
**Contribution:** 2 fair

**Summary:**

This paper proposes to use graph reordering to improve the cache locality of graph-based nearest neighbor search algorithms. An analysis is conducted to show why graph reordering works and the experiments show that graph reordering significantly improves performance.

**Questions:**

See weakness.

**Limitations:**

Yes

**Strengths And Weaknesses:**

Strength

1.	Graph-based nearest neighbor search algorithms are very popular and graph reordering improves the performance of graph-based nearest neighbor search algorithms.

2.	Although the conditions are restricted, the analysis explains why graph reordering improves performance.

3.	The profiling-based ordering scheme makes sense.

Weakness

1.	The authors should dig deeper to show why recording improves performance. Currently, the explanations are vague (e.g., due to software prefetcher and auxiliary functions). The authors may want to make them more specific by explaining how software prefetcher works or conduct some experiments to show where the reduction comes from.

2.	I believe that graph reorder works for graph-based algorithms in general. But it helps to show by experiments that graph reorder improves the performance for algorithms other than HNSW (e.g., NSG or NGT).

3.	The legends in the figures are too small to read.

---

> ### Author Response · Authors · 2022-08-02
> **Better explanation of reordering performance**
>
> Thank you for your review and comments to improve the paper. We address specific issues below.
>
> **Performance via hardware prefetching:** We’ve done some additional experiments to understand where the performance improvements come from. These experiments supplement our brief discussion of prefetching in Section 2 and reinforce our conclusion that hardware prefetching plays a critical role.
>
> First, we turned off the L1/L2 hardware prefetcher to isolate the effect of cache coherency from that of hardware prefetching (see the technical note regarding L3, below). We issued 10K queries and measured the average speedup (listed below). We have also uploaded a figure “new_hw_prefetch.png” to the supplementary ZIP file that shows the full speedup vs. recall tradeoff.
>
> **MNIST (3172 bytes / node)**
> - With HW prefetch: 15%
> - Without HW prefetch: 5%
>
> **DEEP10M (420 bytes / node)**
> - With HW prefetch: 26%
> - Without HW prefetch: 21%
>
> **SIFT10M (164 bytes / node)**
> - With HW prefetch: 25%
> - Without HW prefetch: 20%
>
> Because we see a drop in performance when hardware prefetching is turned off, one of the ways that reordering improves performance is by making it more likely for a next-line or predictive prefetcher to load the correct data vectors. L1/L2 hardware prefetching is actually responsible for a fairly large amount of performance, especially for datasets with large vector sizes (e.g. MNIST). The remaining speedup is likely due to cache coherence for auxiliary data structures such as the list of node statuses and L3 / TLB prefetching (see technical note).
>
> Technical note: We ran the prefetch experiments on an Intel Broadwell Core i5 and we turned off the prefetchers via the MSR registers (specifically, register 0x1A4). This disables the L1 data prefetcher and the next-line prefetchers for the L2 data and instruction caches. It also disables the L2 streaming prefetcher, which can prefetch into either the L2 or L3 cache. However, the BIOS settings expose an LLC prefetch setting for the L3 cache and the TLB cache has its own prefetcher. We were unable to completely disable these for the “No HW Prefetch” results because we do not have unrestricted access to the BIOS / UEFI.
>
> **Performance and vector size:** To understand how data vector size affects reordered performance, we consider two equivalent versions of the SIFT1M dataset - one where vectors are represented as 32-bit floats (512 bytes / vector) and one where vectors are represented as 8-bit integers (128 bytes / vector). We use the same graph for each index and observe that larger vectors result in a smaller speedup (average of 11% vs 19%). See “new_data_vector_sizes.png” in the revised supplementary materials for the full tradeoff.
>
> **Regarding other graph indices:** We agree that NSG and NGT are also good targets for reordering, though we weren't able to integrate with these libraries due to time limitations. Fortunately, a series of recent ablations have shown that these algorithms are all very similar, with few differences that actually affect performance. For example, the hierarchical part of HNSW can be removed completely with no impact on performance (see [1] as well as our ablation in Section 1.2 of the appendix) and NGT's link diversification defaults to that of HNSW under common hyperparameter choices (see Section 1.1 of the appendix). The core drivers of performance seem to be beam search and link pruning [2], which are used with remarkably few differences by the HNSW / NSW family, the NGT / PANNG / ONNG family and the NSG / EFANNA / DiskANN family of graphs.
>
> Finally, thank you for pointing out the figure formatting issue - we will increase the size of all figures in the revision.
>
> ### References
> [1]: Prokhorenkova, L. and Shekhovtsov, A. Graph-based nearest neighbor search: From practice to theory. In International Conference on Machine Learning, pp. 7803–7813. PMLR, 2020.
>
> [2]: Lin, P.-C. and Zhao, W.-L. A comparative study on hierarchical navigable small world graphs. arXiv preprint arXiv:1904.02077, 2019a.

---

### Official Review · Reviewer_Li81 · 2022-07-11

**Rating:** 7
**Confidence:** 4
**Soundness:** 3 good
**Presentation:** 3 good
**Contribution:** 3 good

**Summary:**

The paper studies practical performance of different methods for rearranging the node layout in the memory for graph-based approximate nearest neighbor search algorithms (HNSW specifically). It also proposes a simple modification of the existing methods based on query profiling. The paper claims to have up to 30-50% improvement in query latency on 100M datasets and is accompanied by code (though not clear if it is going to be released).

**Questions:**

I would recommend improving the code presentation and completeness. The approach is focused on hardware optimization, so having an exact working code seems to be essential for the further research from other teams.

It would be nice to check some of the assumptions of the paper, like that the speedup is from the hardware prefetcher. E.g. turning the hardware prefetcher off and/or experimenting with the size of the vectors.

Misc: “Studies show that Studies show that” typo
Please use “nearest neighbor search index” or similar instead of "near neighbor index". "near neighbor index" is used in other contexts, and I was able to find the public version of this paper in google top results just by searching for "near neighbor index" .


**Limitations:**

I do not think there is any negative societal impact.

**Strengths And Weaknesses:**

Strengths:
- Simple change that is very likely to be agnostic to the type of the graph algorithm used.
- Sizable gains from the algorithm on large datasets.
- Source-code (hopefully will be released with the publication - not clear from the text).
- Long discussions of the results.

Weaknesses:
- The paper does not have a clear description of the methods (no pseudocode or even text description of step-by-step actions). I guess the readers are suggested to follow the cited papers, but IMO there should be at least a sketch of the best solution.
- The source of 1000 queries for POrder not discussed in the paper. Were they used from the train set or the test set?
- There is no implementation of POrder in the code, which is confusing. The code also has bugs (e.g. nonexistent “-openmp” flag), it does not compile without fixing the dependencies (could had been fixed if you provide Dockerfile) and there are errors in its description.
- Judging the code, the construction is done in a single thread. If index construction time provided in the paper is for this regime (which is not clear from the paper, but seems to be the case), it should be redone in the multi-threaded regime.

---

> ### Author Response · Authors · 2022-08-02
> **Improved code and new prefetching experiments**
>
> Thank you for your review and for taking the time to look at our code. We address specific issues and questions below.
>
> **Regarding the algorithm listing:** This is a good suggestion, and we will add explicit pseudocode for RCM, Gorder and Porder in the revision.
>
> **Regarding the Porder queries:** We use the first 1000 queries from the test set and report the mean latency over 10k queries.
>
> **Regarding the code:** Thanks for looking at our code - we’re sorry about the dependency / build issues. We’ve updated the supplementary materials with a version that implements POrder and makes many other improvements - see below for details.
>
> **Regarding the construction times:** Yes, everything is single-threaded, including the construction times. We will clarify this in the revision. Our current code does not include parallel construction, but the same locking / batch update algorithm used by HNSW and similar indices applies (note that this is non-trivial to implement correctly). We do want to point out, though, that even if we suppose perfect parallel scaling across 8 threads, reordering is still comparable in cost to index construction.
>
> ## Question Answers:
>
> **Code:** We’ve uploaded a revision of the code that has undergone substantial refactoring to improve the presentation. Specifically, we have:
> - Simplified tools to allow easy benchmarking on ANN-Benchmarks and Big ANN-Benchmarks from NeurIPS 2021
> - Migrated to a cmake build that should address the compilation / dependency issues on most platforms (we’ve tested on a clean install of Ubuntu 20.04.4 and MacOS Monterey 12.3)
> - Included a full implementation of profiled reordering, as well as generally cleaned up the code.
>
> We will release this version with the paper and we hope that it will serve as a good reference implementation.
>
> **Prefetching Experiments:** We were able to run the following experiments to understand how the prefetcher and data vector size affect performance. To isolate the effect of cache coherency from that of hardware prefetching, we turned off the L1/L2 hardware prefetcher (see the technical note regarding L3, below). We issued 10K queries and measured the average speedup (listed below). We have also uploaded a figure “new_hw_prefetch.png” to the supplementary ZIP file that shows the full speedup vs. recall tradeoff.
>
> **MNIST (3172 bytes / node)**
> - With HW prefetch: 15%
> - Without HW prefetch: 5%
>
> **DEEP10M (420 bytes / node)**
> - With HW prefetch: 26%
> - Without HW prefetch: 21%
>
> **SIFT10M (164 bytes / node)**
> - With HW prefetch: 25%
> - Without HW prefetch: 20%
>
> L1/L2 hardware prefetching is responsible for a fairly large amount of performance, especially for datasets with large vector sizes (e.g. MNIST). The remaining speedup is likely due to auxiliary data structures and L3 / TLB prefetching.
>
> Technical note: We ran the prefetch experiments on an Intel Broadwell Core i5 and we turned off the prefetchers via the MSR registers (specifically, register 0x1A4). This disables the L1 data prefetcher and the next-line prefetchers for the L2 data and instruction caches. It also disables the L2 streaming prefetcher, which can prefetch into either the L2 or L3 cache. However, the BIOS settings expose an LLC prefetch setting for the L3 cache and the TLB cache has its own prefetcher. We were unable to completely disable these for the “No HW Prefetch” results because we do not have unrestricted access to the BIOS / UEFI.
>
> **Data vector size:** We consider two equivalent versions of the SIFT1M dataset - one where vectors are represented as 32-bit floats (512 bytes / vector) and one where vectors are represented as 8-bit integers (128 bytes / vector). We use the same graph for each index and observe that larger vectors result in a smaller speedup (average of 11% vs 19%). See “new_data_vector_sizes.png” in the revised supplementary materials for the full tradeoff. These results support the idea that reordering is particularly beneficial when the data vectors are of low dimension / compressed via quantization.
>
> Finally, the revision will fix the typos and naming issues. Thank you for pointing them out.

---

### Official Review · Reviewer_kSKW · 2022-07-12

**Rating:** 6
**Confidence:** 3
**Soundness:** 3 good
**Presentation:** 2 fair
**Contribution:** 3 good

**Summary:**

This paper presents a formal analysis for the impact of graph reordering (i.e., ordering the in-memory storage sequence of graph node embeddings) on the cache efficiency of near neighbour searches using near neighbour graphs. The connection of the graph ordering (i.e., memory layout of the graph nodes) and the cache complexity is formulated, based on which the cache complexity of the Gorder (Wei et al., 2016) is analysed and two other graph ordering (Corder and Porder) are proposed. Experimental results on large real datasets confirm the effectiveness of the analysis and the importance of graph reordering for near neighbour search efficiency.

**Questions:**

A lot of experimental results focused on Gorder (Wei et al., 2016). How will the results look like when Porder is also included in those experiments (Figures. 3&4, Tables 1&3)?

**Limitations:**

The paper included an interesting discussion on the results and limitations.

**Strengths And Weaknesses:**

Strengths:
1. The paper has presented a solid analysis for the impact of graph reordering (i.e., ordering the in-memory storage sequence of graph node embeddings) on the cache efficiency of near neighbour searches using near neighbour graphs.

2. Experimental results on large real datasets are presented, with an interesting discussion on the results.

3. Source code of the paper is available.

Weaknesses:
1. The proposed method Corder is ineffective as discussed in the experimental results. Perhaps this method can be dropped to make room for adding more details on the experimental settings to the main content of the paper.

2. The other proposed method Porder is only somewhat better than the existing method Gorder (Table 5).

3. A lot of contents have been included as supplementary material, which makes the paper somewhat difficult to follow.

Typo: "degree-based groupingFaldu et al. (2019)"; "Studies show that Studies show that"

---

> ### Author Response · Authors · 2022-08-02
> **New benchmarks with Porder**
>
> Thank you for your review - we’re glad you find the problem to be interesting and we appreciate the suggestions to improve the work. See specific points below.
>
> **Regarding Porder across the full recall-latency tradeoff:** We’ve re-run a limited set of experiments on SIFT10M, DEEP10M, GIST1M and DEEP100M. Unfortunately, we no longer have access to the hardware used to obtain Tables 1&3 and Figures 3&4, so the results are not directly comparable with Figure 3&4. We rebuilt all the indices for Gorder, Porder and RCM on a new machine, and we have uploaded a revised version of the supplementary materials that shows the full tradeoff (see “new_porder_benchmarks.png” in the supplementary zip file).
>
> The following speedups should also give a good sense of the trend:
> - SIFT10M at 90% recall: Gorder (3.4%) and Porder (9%)
> - DEEP10M at 90% recall: Gorder (18.4%) and Porder (23.5%)
> - GIST1M at 90% recall: Gorder (8.1%) and Porder (9.5%)
> - DEEP100M at 87% recall: Gorder (29.8%) and Porder (33.1%)
>
> Profiling primarily helps in high-recall applications where the query distribution is non-uniform and the dataset is large. For example in Table 5, we queried at the 99% recall level and profiled with the search parameters that are used to get 99% recall. In the new benchmarks, where we examine a lower recall regime and do not tune, we see smaller speedups - though the method does still produce a better ordering than Gorder. Even though the improvement is small, we still think it is useful, given that Gorder is the best known method and that many real-world search problems are latency-critical with non-uniform, highly skewed query distributions (e.g. in recommender systems, some queries are substantially more common than others and are issued repeatedly).
>
> **Regarding paper organization:** Thank you for the suggestions to improve the organization of the paper. Since the main purpose of Corder is to validate a theoretical claim, we agree that it could be swapped with some experimental details located in the appendix. We plan to add the relevant details from Section 4 of the appendix to Section 6 of the main text and (depending on space limitations) drop the discussion of Corder from that section.

---

### Meta-Review · Area_Chair_ioPC · 2022-08-22

**Recommendation:** Accept
**Confidence:** Certain

**Metareview:**

This paper studies how to order in-memory sequences for graph embedding. There was a positive consensus that the studied problem is interesting and results are sufficiently discussed. There were some concerns on missing results, which were addressed during rebuttals.

**Award:**

No

---

### Decision · Program_Chairs · 2022-09-14

Accept